# Cell geometry regulates tissue fracture

**Amir J. Bidhendi** [1,3] ✉, **Olivier Lampron** [2], **Frédérick P. Gosselin** [2] & **Anja Geitmann** [1] ✉

In vascular plants, the epidermal surfaces of leaves and flower petals often display cells with wavy geometries forming intricate jigsaw puzzle patterns. The prevalence and diversity of these complex epidermal patterns, originating from simple polyhedral progenitor cells, suggest adaptive significance. However, despite multiple efforts to explain the evolutionary drivers behind these geometrical features, compelling validation remains elusive. Employing a multidisciplinary approach that integrates microscopic and macroscopic fracture experiments with computational fracture mechanics, we demonstrate that wavy epidermal cells toughen the plants' protective skin. Through a multiscale framework, we demonstrate that this energy-efficient patterning mechanism is universally applicable for toughening biological and synthetic materials. Our findings reveal a tunable structural-mechanical strategy employed in the microscale design of plants to protect them from deleterious surface fissures while facilitating and strategically directing beneficial ones. These findings hold implications for targeted plant breeding aimed at enhancing resilience in fluctuating environmental conditions. From an engineering perspective, our work highlights the sophisticated design principles the plant kingdom offers to inspire metamaterials.

In plants, anatomy and function are intimately linked and hierarchically structured at the cell, tissue, and organ levels. Since fully differentiated plant cells rarely change shape or location, morphogenesis during tissue development requires the precise choreography of cellular growth and expansion processes[1–4]. Cell shapes in plants are highly diverse and tissue-specific, but for some cell types, the relationship between shape and function has remained elusive. A prominent example is the characteristic pattern formed by pavement cells covering the surface of many eudicotyledon plant leaves (Fig. 1). These tabular cells display undulating borders with geometric complexities that vary across plant families[5]. The pressures on epidermal functioning that may have driven the evolution of this complex, jigsaw puzzle-like cell patterning are not understood. Proposed hypotheses include roles for the undulating cell circumference in conferring elasticity to the tissue under tensile stress[6] or in minimizing surface stress at increasing cell volumes[7]. However, the biological relevance of these concepts is not obvious, and validation for both hypotheses is lacking. Here we explore a novel hypothesis that links the shape of pavement cells to the role of the epidermal tissue as a skin forming the plant interface with the environment.

**Engineering principles governing plant leaf surface architecture**
Green leaves are the primary photosynthetic organs of vascular plants and their characteristic flat shape maximizes light capture and optimizes gas exchange[8]. This optimization of organ shape for photosynthesis comes at an ecological cost, however—high exposure to and susceptibility to mechanical damage by biotic and abiotic factors such as herbivory, pathogens, hail, sand storms and wind[9–12]. The abrasive, piercing or slicing actions of these agents can initiate holes in the leaf surface that are prone to propagating as cracks. Where the epidermis is intact, it endows the plant leaf with a resilient and hydrophobic surface whose only openings are controllable valves—the stomata (Fig. 1C). Physical damage to the leaf surface affords pathogens ready access to internal tissues[12] and exposes the photosynthetic mesophyll to uncontrolled dehydration. Even microscopic damage to the surface poses a high risk to plant health and survival. We propose that epidermal

[1]Department of Plant Science, McGill University, Macdonald Campus, 21111 Lakeshore, Ste-Anne-de-Bellevue, Québec H9X 3V9, Canada. [2]Laboratoire de Mécanique Multi-échelles, Département de génie mécanique, École Polytechnique de Montréal, Montreal, Québec H3C 3A7, Canada. [3]Present address: EERS Global Technologies, Montreal, Canada. ✉e-mail: amir@bidhendi.net; geitmann.aes@mcgill.ca

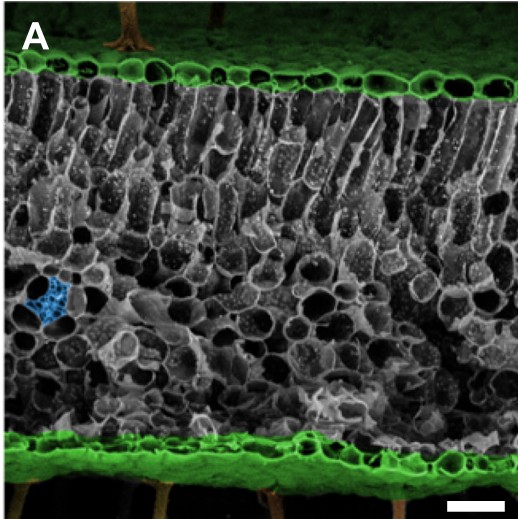

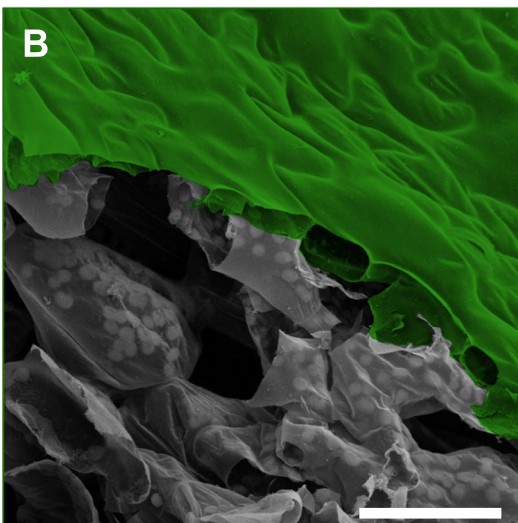

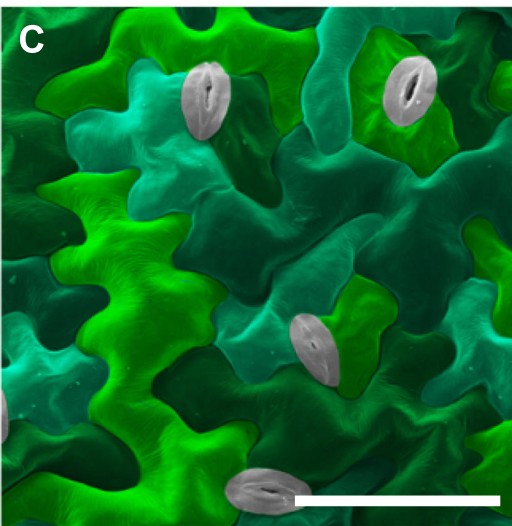

**Fig. 1 | Scanning electron micrographs of plant leaf structures. A** Cross-section of *Geranium sp.* leaf showing epidermal layers (colorized green) sandwiching layers of mesophyll cells (gray) and vascular tissue (blue). **B** Cutaway of *Arabidopsis thaliana* leaf demonstrating the epidermis (green) and underlying mesophyll cells (gray). Note the tight adherence between pavement cells compared with the large air spaces between the mesophyll cells. **C** Orthogonal view of pavement cells at the surface of an *Arabidopsis thaliana* cotyledon. Pavement cells colorized in green/teal, stomata in gray. Micrographs exemplify cell organization and patterning that are widely found throughout the plant kingdom. Scale bars = 50 μm.

architecture is crucial for the leaf's surface toughness at the microscopic level, thereby protecting it from potentially fatal consequences of physical damage to its skin. We hypothesize that the jigsaw puzzle cell shape in the eudicotyledon epidermis (Fig. 1C) enhances the tissue's resistance against microcrack propagation. We propose that this cell shape functions to mitigate the risk of growing microfissures on the plant's surface, which could shorten the life span of its vital organs.

## Results

### Geometry influences fracture propagation

Human-made microstructured materials can be designed for increased toughness by optimizing topology to enhance crack deflection or promote crack meandering[13]. Similar toughening mechanisms are also prevalent in natural materials such as skin, bone and nut shells[14–17]. Making the crack path complex at the microscopic level increases the required work and thus retards the propagation of a crack across a macroscopic distance. Structural features that promote crack deflection are, for example, interfaces between dissimilar materials[18–20]. We hypothesize that the undulating shape and jigsaw puzzle-like arrangement of eudicotyledon pavement cells increase microscopic epidermal toughness by creating irregularly oriented interfaces that deflect crack paths and force growing fissures to meander. To test this hypothesis, we created a macroscopic physical model of epidermis, by patterning sheets of cast polymethylmethacrylate (PMMA), commonly known as acrylic or Plexiglass, with the shapes of epidermal pavement cells. Using a physical model allowed us to disentangle two factors that potentially influence fracture paths in plant material: the structure (geometry of interfaces) and the material properties of the cell walls. The plant cell wall is a composite material with a fiber component typically arranged in a non-random fashion, conferring anisotropic material behavior[21–25]. The use of isotropic PMMA sheets eliminated this confounding variable, allowing us to focus solely on the effect of cell border geometry, which we implemented in the PMMA sheets via laser engraving (Fig. 2, Supplementary Fig. 1). The effect of wavy jigsaw puzzle cell patterns adopted from a true *Arabidopsis* leaf was compared to that of simple brick-shaped cells characteristic of onion leaf epidermis. A non-engraved sheet served as control. Application of stress on the laser-engraved specimens demonstrated cracks to alternate between progressing along the outline of the cells ('cell interfaces') and traversing 'cells'. Crack paths that traversed the brick-shaped onion cell pattern behaved differently depending on their orientation relative to the cell's long axes. A crack initiated in parallel to the cells' long axes traveled long distances along cell interfaces (Fig. 2A), whereas a crack initiated perpendicular to the onion cells frequently alternated between following cell interfaces and traversing cells (Fig. 2B). Similarly, wavy cell interfaces frequently deviated propagating cracks (Fig. 2C, D). The work-to-fracture values for specimens with wavy and brick-shaped interfaces, when oriented perpendicular to the crack path, were higher not only than in samples with brick-shaped interfaces oriented along the crack path but also higher than in unpatterned (control) samples (Fig. 2E, F). In other words, the engraving patterns toughened the PMMA sheets.

### Numerical simulation of crack propagation

To further refine the conclusions made based on the engraved PMMA material, we simulated crack propagation in silico using a variational phase-field fracture model (PFM, see Methods) which is derived from the reformulation and regularization of Griffith's theory for brittle fracture[26,27]. PFM is a computational method used for simulating the formation and evolution of complex fractures in a material. Methods developed earlier[28] allowed us to predict crack initiation, propagation, and coalescence without the need for ad hoc criteria and eliminating challenges common to conventional approaches such as linear elastic fracture mechanics (LEFMs) and cohesive zone models. We assumed that the cells are homogeneous regions with an isotropic elastic and

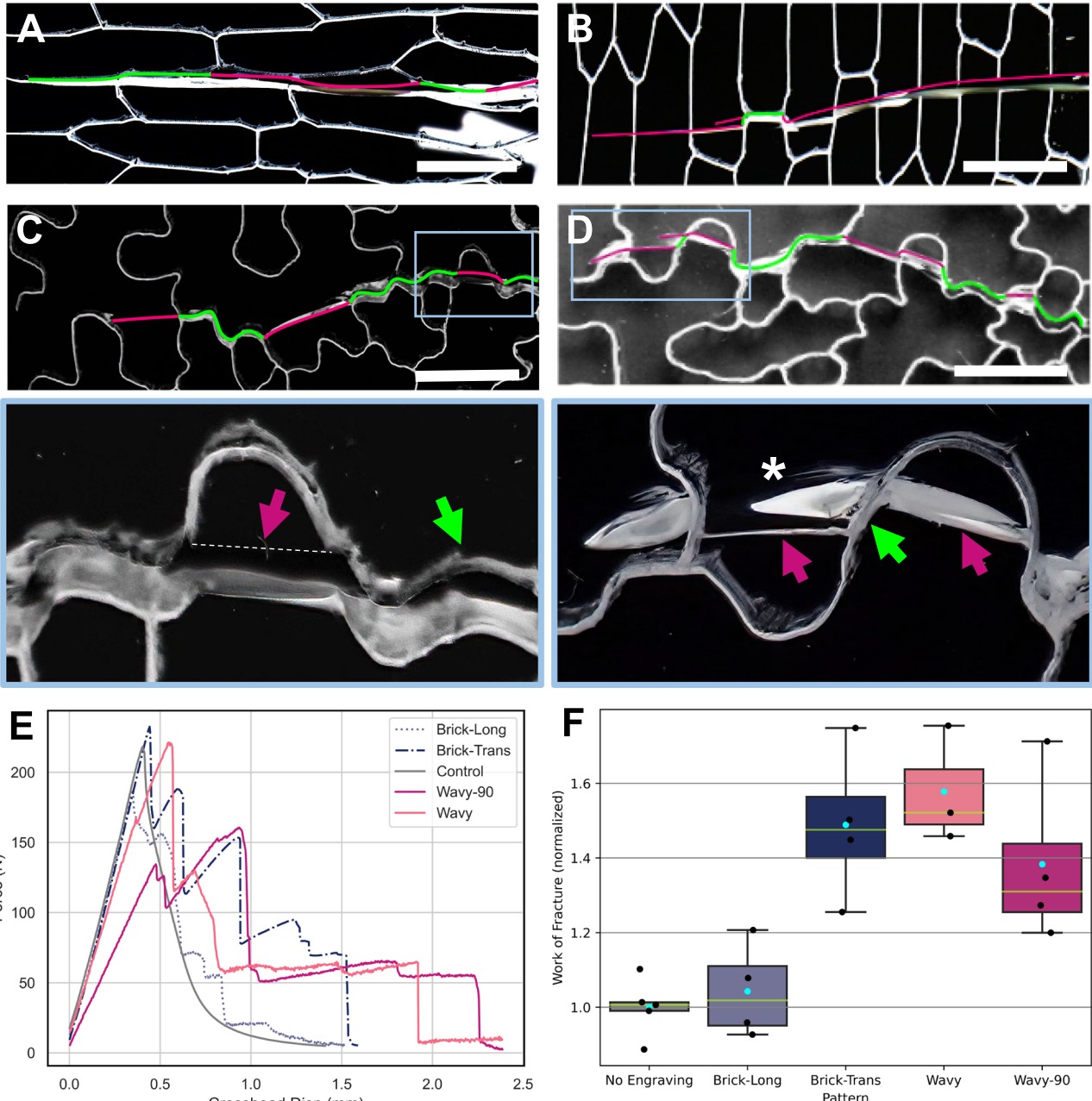

**Fig. 2 | Fracture behavior of patterned PMMA sheets.** Specimens featuring brick-shaped cell patterns with crack initiated along (**A**) and perpendicular (**B**) to the cells' long axes. Cracks take approximately straight paths - following cell interfaces if on the crack path (compare green and magenta portions). **C, D** Cracks in the wavy cell patterned samples take rugged paths following both engraved borders (green) and crossing cells (magenta). Shallow undulations cause the crack to follow interfaces (green) leading to 'cell-cell separation' while deeper undulations force cracks to alternate frequently between 'interface' and 'cell' (magenta). The asterisk marks an arrested crack with a bifurcation which follows the interface briefly before entering the cell. **E** Force profiles during crack propagation through engraved PMMA sheets. The control (no engraving) experiences a smooth brittle fracture, similar to the onion-patterned samples cracking along cell alignments (brick-long).

All other samples continue to bear load after the onset of crack propagation featuring several local maxima. Wavy and Wavy-90 refer to a pavement cell pattern and its 90-degree rotation. **F** Boxplots of Work-of-Fracture in PMMA sheets (normalized to control specimens) based on engraving pattern with $n = 5, 4, 4, 3$ and 4 independent samples used for control, longitudinal and transverse brick-shaped, wavy, and its 90-degree rotation patterns, respectively. The central rectangle signifies the Interquartile Range (IQR), with the upper edge denoting the Third Quartile (Q3) and the lower edge representing the First Quartile (Q1). Horizontal green line marks the median, cyan dots indicate the mean, black dots are individual data points. Whiskers extend to 1.5 times the IQR above Q3 and below Q1. Scale bars = 1 cm.

brittle fracture behavior, joined together by a thin second phase which corresponds to the engraved cell interfaces. 'Cells' and 'interfaces' were formulated similarly but were assigned different material properties (see Methods). The numerical models were based on 2D patterns mirroring the geometry, dimensions, and patterning of the acrylic compact-tension specimens.

To investigate the effect of interface geometry on the sample's effective toughness, interfaces were given the same elastic modulus as cells, but a lower toughness ($G_c^{int} = G_c^{cell}/2$). Samples with brick-shaped cells aligned parallel to the direction of crack propagation demonstrated a lowered resistance compared to the control while the toughness of the other samples remained close to the control

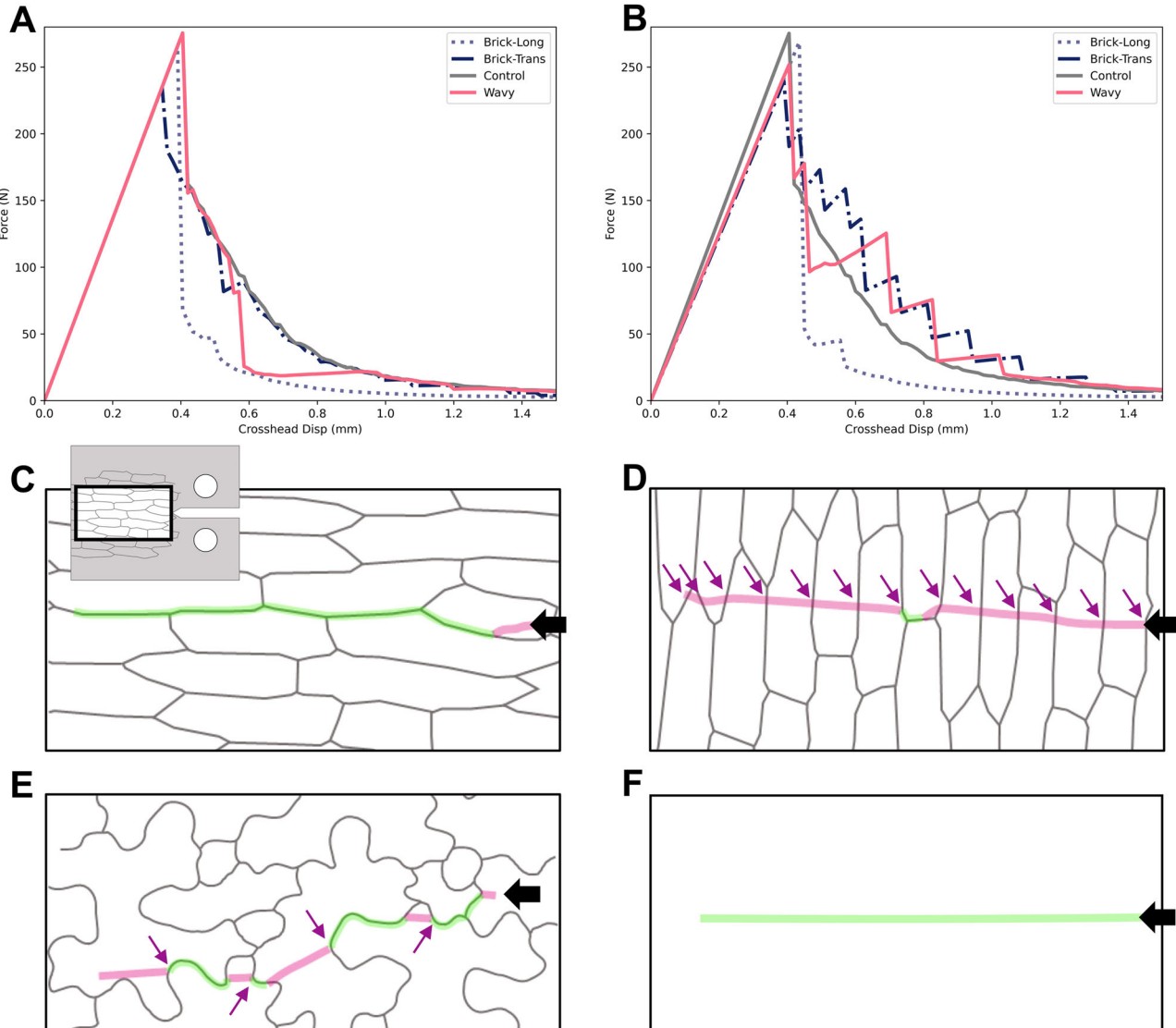

**Fig. 3 | Numerical simulation of crack propagation in patterned material using phase-field fracture model. A** Force-displacement curves obtained for control and patterned specimens for $E^{int} = E^{cell}$ and $G_c^{int} = G_c^{cell}/2$. The brick-shaped cell patterns oriented parallel to the initiated crack exhibited dramatic brittle behavior as crack propagation followed interfaces (see Methods). The transversely oriented brick-shaped pattern and the wavy cell patterns showed similar force-displacement curves to the control profile. **B** Force-displacement curves for $E^{int} = E^{cell}/2$ and $G_c^{int} = G_c^{cell}/2$. Brick-shaped cell patterns with longitudinal orientation were brittle

as in (**A**), but the other specimens showed an increased toughness compared to the control. Closeup views of crack paths for longitudinally and transversely aligned brick-shaped cells (**C**, **D**) as well as wavy cell patterns (**E**) and control specimen (**F**, no engraving). Compare green and magenta segments of crack path marking border and cell wall fracture portions, respectively. Black arrows indicate the direction of initial crack. Purple arrows indicate instances of crack re-initiation when the propagating fracture enters stiffer cells from softer interfaces. See also Supplementary Fig. 4.

specimen (Fig. 3A, Supplementary Fig. 2). Next, interfaces were given both lower toughness and lower elastic modulus. Parameters for the interface material and cells were set to be identical, except for toughness $G_c^{int} = G_c^{cell}/2$ and Young's modulus $E^{int} = E^{cell}/2$ (Fig. 3B). Similar to the results obtained with the PMMA sheets, the force-displacement curves while cracking a computational sample with brick cells arranged longitudinally were smooth because the crack propagated mostly along the interfaces (Fig. 3C), and as the crack propagated, the force values dropped more rapidly than in the control specimen (Fig. 3B). With cells arranged transversely (Fig. 3D), however, the force-to-fracture values oscillated in a decreasing sawtooth fashion (Fig. 3B). Several distinct mechanisms have been identified to potentially underlie such a phenomenon in heterogeneous materials[29], and here we suggest that it resulted from a difference in Young's modulus between the two phases (cells and interfaces), forcing the crack to re-initiate each time it transitioned from a weaker interface to

a stiffer cell[30]. Each re-initiation of a crack required a temporary increase in load for the crack tip to accumulate the necessary strain energy. The same phenomenon was seen with the wavy *Arabidopsis* cell pattern with even more pronounced crack meandering (Fig. 3B, E). It is remarkable and counterintuitive that as long as interfaces were oriented in directions other than parallel to the crack propagation direction, the inclusion of a weaker interface material augmented the effective toughness of the specimen. The extent of the effect on toughness was highly dependent on cell geometry and the resulting magnitude of crack path deviation. Our data highlight the significance of interfacial properties in influencing fracture behavior.

## Crack behavior in the plant epidermis validates deflection and meandering

Based on the macroscopic crack behavior of patterned PMMA material and the numerical models, we predicted that in jigsaw puzzle

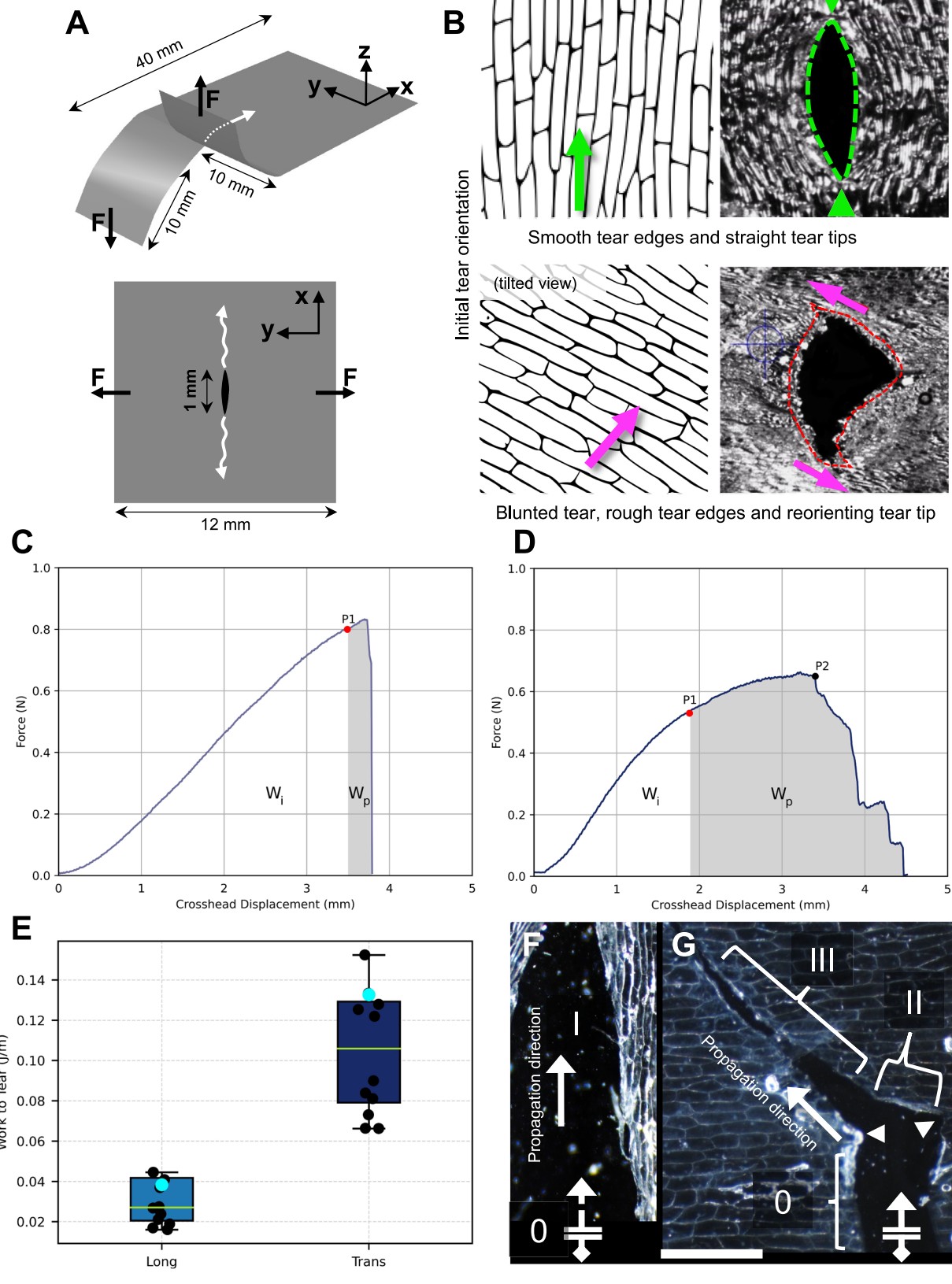

patterned leaf epidermis tissue, fractures propagate alternatingly along cell interfaces and traversing cells. In a typical eudicotyledon epidermis, the resulting zigzag pattern is expected to cause a meandering crack. In a typical monocotyledon epidermis with straight cell borders, on the other hand, cracks are predicted to propagate mostly straight when initiated parallel to the principal cell axes and to

meander when the crack is initiated perpendicularly to the cell long axis.

To validate these predictions in biological specimens, excised epidermal tissues from onion leaves were submitted to mechanical testing. *Arabidopsis* epidermis proved too delicate to be isolated intact from the underlying mesophyll layers to be tested similarly.

**Fig. 4 | Tear testing of onion epidermis. A** Geometry and loading of tear tests. Top: Two-leg trouser tear test, Bottom: Notched specimen tear. F denotes the direction of force applied with respect to the xy plane. The orientation of the original blade-cut notch is along x. The two tests differ in the placement of the initial notch and in the orientation of load application (perpendicular to the xy plane for two-leg trouser tear test). **B** Tearing of epidermis with a notch in the center. Top: tear along the cells' long axes features a smooth path. Bottom: Tear transverse to cells' long axes demonstrates that while the path has the global tendency to proceed according to the applied force field, minute, local deviations are induced through microscopic reorientations imposed by cell-cell interface features, leading to rough tear edges and blunted tear tips (see also Supplementary Movies 1,2). Micrographs are representative of over twenty observations. **C**, **D** Force displacement profiles of specimens stretched in (**A**) and (**B**), respectively. P1 marks the onset of tear growth. P2 marks the peak force after which the tear turned slightly toward the cell direction leading to a drop in force, although the tear did not freely propagate in this direction. $W_i$ and $W_p$ refer to Work for initiation and propagation of tear, respectively. The tear propagation zone for longitudinally-notched specimens was negligible (brittle behavior). **E** Normalized Work-to-Tear for transversely- and longitudinally-notched specimens with $n = 12$ and 16 independent samples used in analysis for longitudinal and transverse samples, respectively. The central rectangle signifies the Interquartile Range (IQR), with the upper edge denoting the Third Quartile (Q3) and the lower edge representing the First Quartile (Q1). Horizontal green line marks the median, cyan dots indicate the mean, black dots are individual data points. Whiskers extend to 1.5 times the IQR above Q3 and below Q1. Data points below Q1-1.5*IQR or above Q3 + 1.5*IQR were classified as outliers (not shown). Tearing of two-leg trouser specimens parallel (**F**) and transverse (**G**) to cell orientation. While along the cells the tear path remained relatively straight, it showed a marked difficulty crossing from one cell to the next. Type 0 tear path corresponds to the location of the initial notch. Type I marks a tear path parallel to cell lines. Type II is a 90-degree reorientation of the tear path and type III makes a subsequent reorientation of the tear path towards an oblique angle. Different tissue samples showed all or some of these tear path types in varying order. Tears driven perpendicular to cell interfaces were frequently associated with reorientation and rugged edges.

We performed two-leg trouser and notched tear tests (Fig. 4A) on onion epidermis using a custom-built device for mechanical testing of delicate plant materials[31,32]. The two-leg trousers tear test is a mechanical test setup where the sample is shaped like a pair of trousers with a pre-cut to initiate a tear; to measure tear resistance, the 'legs' are pulled apart in the same direction as the leg motions of a walking person. The notched tear test involves a pre-cut or notch in the sample, subjected to tensile force to assess resistance to tear propagation. Both tests are used to analyze fracture mechanics and tear resistance. For notch tests, a small slit, serving as starting point for the tear, was cut in the center of the sample (Fig. 4B). This approach stabilizes the tear path to propagate across the width of the sample (see Supplementary Note 1). Consistent with the PMMA fracture results, the work-to-tear in the onion epidermis transverse to cell orientation was significantly higher (~350%) than along their orientation (Fig. 4B–E, $p < 0.05$. See also Supplementary Movies 1,2). Tears in the two-leg trouser samples tended to propagate more freely than those in the center-notched test samples. Tears that originated parallel to the longitudinal onion cells followed a relatively straight path (Fig. 4F, Supplementary Movie 3, Supplementary Fig. 3), while those initiated perpendicular to cell orientation were typically reoriented (Fig. 4G, Supplementary Movie 4). Reorientation occurred either towards the long axis of the cells (type I) or towards an oblique angle (type II) suggesting that both cell interfaces and material anisotropy of the biological samples (absent in isotropic PMMA) influence tear behavior. We speculate that the oblique tear orientation may be a result of the oblique orientation of cellulose microfibril bundles (Supplementary Fig. 3D) in onion epidermal cells[33–36]. The observed reorientation of tears confirms a variable resistance in the onion epidermis, which shifts from sustained to abrupt failure depending on the tear path's alignment with the cell orientation. These experimental observations align closely with our findings from patterned PMMA described earlier and with numerical simulations, validating our models in their representation of tear propagation in leaf epidermis.

**Crack behavior at cell interfaces**
Finally, we wanted to examine where exactly tears occur and how they propagate at the subcellular level. The 3D structure of a plant epidermal cell consists of two parallel periclinal walls forming the 'floor' and 'ceiling' surfaces connected by orthogonal anticlinal walls. A cell-cell interface consists of the two adjacent anticlinal walls of neighboring cells, adhered together by the very thin middle lamella (Fig. 5). When approaching an interface, a propagating crack, therefore, encounters multiple structural and material discontinuities.

We asked whether tearing proceeds along the inside edge of the cell-cell interface to avoid traversing the anticlinal wall, or whether tears would preferentially travel between cells, separating them at the middle lamella. In the cotyledons of *Arabidopsis* embryos, the tearing force was observed to separate the cells at the middle lamella[37] (Fig. 6A, B), suggesting that in these young tissues, the middle lamella has not stiffened sufficiently to prevent cell dissociation under stress. It was not possible to determine whether the splitting at the middle lamella was due to adhesive failure, which is a rupture at the interface between the bonding material and substrate, or cohesive failure, the rupture of the bonding material itself (Fig. 5B). On the other hand, in the mature tissues of the tomato and onion leaf epidermis, a tear path either crossed the cells (as in Fig. 5C) or followed interfaces at the inside edge of the cell border (as in Fig. 5A), whereas only occasionally it entered the space between the anticlinal walls (as in Fig. 5B) leading to their detachment (Fig. 6C–E). This behavior was true for both brick-shaped and jigsaw puzzle-shaped cells (see also Supplementary Movies 3, 4). As a result of a strong middle lamella in mature tissues, a crack was readily guided to proceed through the center of the cell, especially when combined with meandering undulating borders. This explains the low resistance to crack propagation along brick-shaped cells. Tears running along straight cell borders were observed to propagate rapidly and at lower forces (Supplementary Movie 3). Treatment of the epidermal tissues with boiling or sodium hypochlorite (Fig. 6F–H) resulted in a higher number of crack paths separating cells at the middle lamella suggesting that the material composing the middle lamella is more readily weakened by elevated temperature or chemical modification than the cell wall material. From these observations, it emerges that both the modulation of cell-cell adhesions as well as geometrical cell patterning constitute tuning parameters that determine tissue integrity.

## Discussion
Over recent decades, substantial research has focused on 'how' pavement cells develop their wavy geometries and 'why' these forms exist. Much progress has been made regarding the 'how'–the developmental and mechanical mechanisms behind these shapes–through the investigation of biomechanical processes and the analyses of molecular players[2,3,7,38–52]. The 'why'–the physiological or evolutionary benefits–on the other hand, remains poorly understood. Existing hypotheses suggest these geometries could either increase tissue elasticity or minimize stress within larger epidermal cells, but these notions have remained speculative[6,7]. Here, a combination of macroscopic and microscopic experimental assays and numerical simulations reveals that wavy cell shapes equip the leaf lamina with an intricate tunable mechanism that reduces its susceptibility to surface damage. We discovered that rather than requiring costly and heavy building material, geometrical patterning of cells endows the leaf with mechanical strength and the ability to hamper the propagation of detrimental surface fissures–an important feature that minimizes the

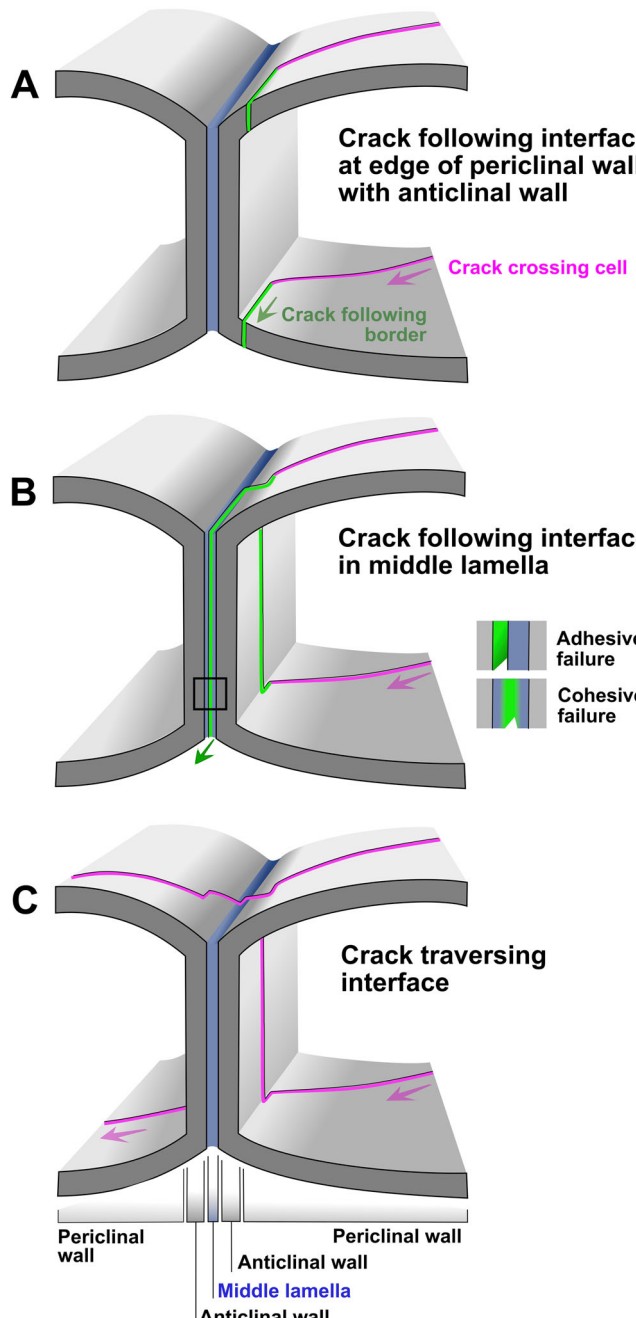

**Fig. 5 | Crack propagation at cell-cell interfaces. A, B** A crack traversing a cell (magenta) and arriving at a cell-cell interface is deviated to propagate along the interface (green). In (**A**) this occurs through a failure in the periclinal wall at the edge towards the anticlinal wall. In (**B**), the crack traverses an anticlinal wall and failure occurs in the middle lamella, either through adhesive failure (between the middle lamella and anticlinal wall) or cohesive failure (in the middle lamella material). **C** Crack traverses the cell-cell interface along a path that remains globally straight but locally shows minor reorientations due to changing material properties when traversing the anticlinal wall, middle lamella, and second anticlinal wall.

risk of dehydration or pathogen invasion thus optimizing the life span of the organ. These insights into the role of cell geometry also have implications for the guidance of beneficial cracks that serve plant survival or propagation. At certain development stages, fractures are required, e.g., upon fruit opening for seed dispersal, or when large leaves split length-wise to reduce fatal drag from high winds. The strategic guidance of crack propagation by way of cell shape may reveal further details on how geometrical features rather than costly

metabolic processes are employed for defense and survival purposes. This concept of resource-efficient structural robustness has the potential to inform breeding efforts aimed at increasing crop resilience against biotic and abiotic environmental stress factors. In addition, the insights gained enrich the knowledge base of natural processes and structural features that can be leveraged for bio-inspired design principles in material science.

## Methods

### Plant materials

Seeds of *Arabidopsis thaliana* Col-0 wild-type (Arabidopsis Biological Resource Center, identifier CS70000) and *anisotropy1 (any1)* (obtained from Geoffrey Wasteneys, University of British Columbia, Canada) were germinated in sterile Petri plates containing 1/2x MS[53] media under long-day lighting condition. For tests on true leaves, the seedlings were transplanted one week after germination and placed in growth chambers until the experiment. Fresh white onions (*Allium cepa*) were obtained from a local supermarket. Adaxial or abaxial epidermal layers were excised from scales 2, 3 and 4, counting from the most external layers, similar to Tirichine et al. (2009)[54]. Most experiments were performed on the adaxial epidermis since it separates easily from the underlying mesophyll leaving epidermal cells intact. Peeling of the abaxial epidermis tends to rip the cell layer in half allowing for experiments that focus exclusively on cell wall material. Other plant types used in the study were obtained from on-campus greenhouses.

### Plant sample preparation

**Specimens for mechanical testing.** For true leaves of *Arabidopsis* or tomato, tissue segments were excised avoiding central veins. For the tear test of the edge-notched sample, epidermal segments were cut to be 40 mm long and 20 mm wide. A 10 mm cut into the sample was made in the middle of the shorter edge, parallel to the long edge. For center-notched specimens, the sample gauge dimensions were $12 \times 12$ mm. A 1 mm long slit was punched in the center of the sample with a 1-mm blade (Fig. 4A). For consistency, samples were prepared using custom cookie cutter-style blades designed for this purpose. During tests involving fresh samples, double-distilled water (ddH$_2$O) was layered on the sample surface and refreshed as needed using a pipette to prevent sample dehydration.

**Sample preparation for scanning electron microscopy (SEM).** Samples were fixed using either formaldehyde or methanol-ethanol fixation methods. For formaldehyde fixation, samples were placed in 3.5% formaldehyde, freshly prepared in PBS buffer with pH 7.3 similar to Tirichine et al. (2009)[54], for 3 h at room temperature. The samples were then rinsed 3 times thoroughly with PBS buffer. Samples were then dehydrated in a graded ethanol series of 20%, 50%, 70%, 80%, and 90%, for 20 min each, followed by a 2-h submersion in 95% ethanol and 3 times submersion in 100% ethanol for 30 min each. After ethanol dehydration, the samples were critical point dried using a Leica EM CPD300 and gold-palladium coated under a Leica EM ACE200 before observation under vacuum in the SEM. As an alternative to formaldehyde fixation, a methanol-ethanol fixation procedure was adopted similar to Talbot et al. (2013)[55]. In this process, samples were first submerged in methanol for 3 h before immediate transfer and submersion in 100% ethanol for 4 h. For large or thick specimens, the samples were left overnight in fresh 100% ethanol. The samples were then dehydrated by critical point drying.

### Microscopy

FEI Quanta and tabletop microscope Hitachi TM-1000 SEMs were used to observe fixed and dehydrated samples. Tensile tests and real-time microscopy of tear propagation in the onion epidermis were carried out under a Zeiss Discovery V8 stereomicroscope. To visualize cell

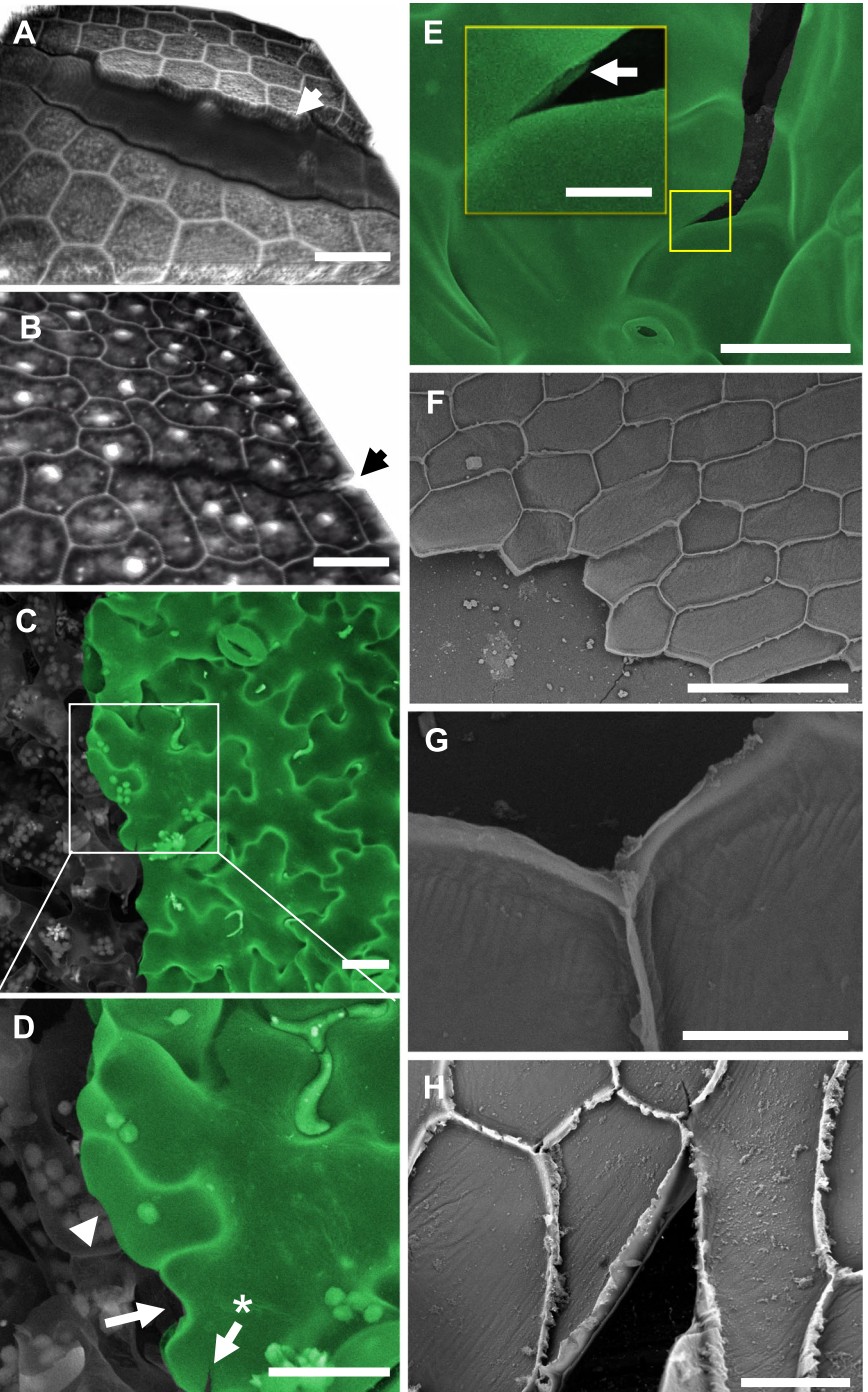

**Fig. 6 | Crack propagation in the plant epidermis.** 3D reconstruction of confocal micrographs of cracks in (**A**) radicle and (**B**) cotyledon epidermal tissues of an *Arabidopsis* embryo extracted by removing the seed coat. The embryo was squeezed between the glass slide and the coverslip. Cell-cell detachment can be observed. Cell wall staining was performed using propidium iodide, which at early growth stages can also label nuclei. Fracture at the middle lamella (as in Fig. 5B) rather than at the inside edge (as in Fig. 5A), was confirmed by the observation of the presence of anticlinal walls on both sides of the crack and by the fact that cells had retained their nuclei indicating that the protoplasts were intact. **C** Scanning electron micrograph of fractured tomato leaves demonstrated a meandering crack path. **D** Close-up of the box in (**C**). The arrow points to a shallower lobe with an interface fracture (as in Fig. 5A). Arrow with asterisk demonstrates an arrested crack

bifurcation traversing the cell (as in Fig. 5C). Arrowhead points to a crack traversing a cell. **E** Close-up of a local crack in *Arabidopsis* leaf induced during application of tensile stress. Crack in the cell wall seems to propagate by splitting the middle lamella (as in Fig. 5B) and the arrow points to a partial view of an anticlinal wall. **F**–**H** Outer periclinal walls of abaxial onion epidermis split open upon peeling. Brief exposure to boiling water (**F**, **G**) or bleach (**H**) prior to peeling dramatically weakened cell-cell adhesion. As a result, tearing the excised epidermal tissue perpendicular to the cell axes facilitated failure at the middle lamella (as in Fig. 5B). Micrographs in Fig. 6 are representative of approximately 350 observations of fractures in plant epidermal tissues. Scale bars = 20 μm (**A**–**D**), inset in (**E**) = 5 μm, 200 μm (**F**), 30 μm (**G**), 50 μm (**H**).

borders (Fig. 6A, B), samples were incubated with 250–500 µg/mL of propidium iodide for 15 min at room temperature. The samples were then thoroughly rinsed with ddH$_2$O before observation on a Zeiss LSM 510 META confocal laser scanning microscope with excitation wavelength of 532 nm and bandpass emission filter of 550–615 nm. To study the organization of cellulose microfibril bundles in onion epidermal cells, the epidermal patches were labeled using 0.5% Pontamine Fast Scarlet 4B for one hour and visualized using a Zeiss Axio observer Z1 spinning disk confocal microscope with excitation wavelength of 561 nm.

### Image processing

General analysis of micrographs was carried out using ImageJ[56]. 3D reconstruction of Z-stacks for visualization of cracks in *Arabidopsis* embryo (Fig. 6A, B) was performed using Amira software (Visage Imaging). Colorizing of SEM micrographs was performed using GIMP (GNU Image Manipulation Program, Gimp.org) which was also used to remove the background of some micrographs to facilitate interpretation.

To extract the pattern of cell borders for laser engraving, micrographs of pavement cells of *Arabidopsis* and onion epidermal cells were obtained through confocal and stereo microscopy using fluorescence labeling as described in the previous section and in Bidhendi et al. (2020)[57]. Images were then imported into the open-source Inkscape software (Inkscape.org) where cell outlines were vectorized to produce input files for the laser cutter.

### Mechanical testing of biological samples

Preliminary tensile and fracture tests were carried out on Liveco Vitrodyne V-200, a miniaturized tensile testing setup developed by Lynch and Lintilhac (1997)[58]. Complementary experiments were carried out on a mechanical testing setup we developed in Bidhendi et al. (2020)[31] that allowed the real-time observation of the mechanical behavior of tissue under a stereomicroscope[32]. Briefly, the custom-built tensile device allows for submicron displacement resolution and the use of a wide range of force sensors to match the force sensitivity required for the samples. The force-displacement data are used to calculate stress-strain graphs. For details and results of tensile tests performed to evaluate the stiffness anisotropy of the adaxial onion epidermis refer to Bidhendi et al. (2020)[32]. To study crack propagation in wildtype and *any1 Arabidopsis* leaves, samples were stretched to failure at speeds of 25, 125, and 250 µm/s. Edge-notched and center-notched onion epidermis specimens were stretched at speeds of 100 and 200 µm/s, respectively. Because of artifacts obscuring the crack path in fresh leaves, we performed edge-notched tear tests on dehydrated leaves (see Supplementary Note 1). In this case, due to the fragile nature of the samples, we performed the tear tests manually by pulling apart the two legs using tweezers.

### Laser engraving and fracture test of Polymethylmethacrylate (PMMA) engraved physical models

Wavy pavement cell patterns were obtained from confocal micrographs as detailed in the previous sections. Onion epidermal cell patterns were obtained from optical micrographs obtained using a stereomicroscope. These were scaled up by a ratio of 54:1 and 250:1, respectively. Cast PMMA was used to produce physical models for its isotropic properties and its proven performance in laser engraving applications. The thickness of the PMMA sheet was 5.58 mm (0.22 in.) and the engraving was performed to a depth of 2.5 mm. A Trotec Speedy 300 laser engraver was used to engrave the patterns and to cut the compact tension (CT) samples. For engraving, the device was set at 100% and 10% of maximum laser power and speed, respectively. The frequency and the resolution were 1000 PPI and 600 dpi, respectively, and each line was passed over 3 times. The CT sample dimensions were adopted from ASTM D5045-99 and modified to accommodate the engraving patterns (Supplementary Fig. 1). After engraving and cutting, as is common in CT fracture toughness tests, a microcrack was induced at the tip of the laser-cut notch by gentle tapping using a snap-off blade knife (Supplementary Fig. 1B inset). The fracture tests of acrylic CT samples were carried out on an MTS Insight machine at a jaw separation speed of 4 µm/s (Fig. 2 and Supplementary Fig. 1).

### Data analysis

Data analysis and visualization were performed using Python programming language, utilizing Pandas, Matplotlib and Seaborn for data manipulation and generating plots. Statistical analyses were conducted using the scipy.stats library in Python. Measurements were obtained from distinct samples and the sample was discarded afterwards. Work of fracture (WOF) of samples, either onion epidermis or PMMA CT samples, was calculated by integrating the area under the curve (AUC) of force-displacement. For biological tissues, the sample size was ≥15 for each group. Overlaid strip plots for each group are presented in Fig. 4E. Outliers, identified based on a 1.5 interquartile range (IQR) criterion, were included in the calculation of medians in the plots but excluded from the visual representation for the sake of plot readability. Their presence in the datasets affected the normality of the distributions; therefore, two-sided Mann-Whitney U test was employed for statistical analysis. The AUCs for center-notched onion samples torn along and perpendicular to cell axis were statistically different ($p = 0.0004$, Mann–Whitney $U$ test). Because the force magnitudes and trends obtained in the fracture test of PMMA samples were observed to be highly consistent, a sample size of $3 < n < 5$ was deemed sufficient for each engraving pattern group. Individual data points for each group are shown in overlaid strip plots provided in Fig. 2F. Five data sets were analyzed: control (no engraving) and patterns of onion cells placed longitudinally (Brick-Long), transversely (Brick-Trans), pavement cell pattern and its 90-degree rotation (Wavy). Statistical analyses were performed using a two-sided Mann–Whitney $U$ test on data normalized by the mean of the "No Engraving" control group. The "Brick-Long" group showed no significant difference with a $p$-value of 0.905. In contrast, the "Brick-Trans," "Wavy," and "Wavy-90" groups had $p$-values of 0.016, 0.036, and 0.016, respectively. Given the small sample size for PMMA fracture in each group, the exact effect sizes were not analyzed.

**Phase-field modeling and simulation of crack propagation.** To simulate crack propagation we used a variational PFM that has emerged from the reformulation and regularization of Griffith's theory for brittle fracture[26,27]. Consider a brittle elastic body $\Omega \in \mathbb{R}^\delta (\delta = 2,3)$ with an already existing crack $\Gamma_c \subset \Omega$. We define the displacement field $\boldsymbol{u}: \Omega \backslash \Gamma_c \rightarrow \mathbb{R}^\delta$ and the potential energy of the body $\Pi$ with

$$\Pi = \int_{\Omega \backslash \Gamma_c} \Psi(\boldsymbol{u}) \mathrm{d}\Omega - \int_{\Gamma_N} \boldsymbol{t} \cdot \boldsymbol{u} \mathrm{d}s, \qquad (1)$$

where $\Psi$ is the elastic strain energy and $\boldsymbol{t}$ is the traction load applied to the $\Gamma_N$ portion of the body's boundary $\partial\Omega$. Griffith's theory for brittle fracture[59] relies on the association of energy to the creation of a crack surface. Therefore, the total energy of the cracked body $E$ can be written as

$$E = \Pi + W_c, \qquad (2)$$

where $W_c$ is the energy associated with the crack surface. Applying equilibrium to the variation of the total energy for an infinitesimal crack extension yields the equality

$$\frac{\partial \Pi}{\partial \Gamma_c} = -\frac{\partial W_c}{\partial \Gamma_c}, \qquad (3)$$

where we can define the energy release rate $G = \frac{\partial \Pi}{\partial \Gamma_c}$. Introducing the notion of a critical energy release rate $G_c$, representing the energy required to create a unit crack surface in the material, Griffith's criterion for propagation can be simply stated as

$$G \geq G_c, \tag{4}$$

Equation 4 implies that a crack will only propagate if its extension releases equal or greater elastic energy than the energy required for the creation of the crack surfaces[60].

Although the cornerstone of LEFM, Griffith's theory can be difficult to apply to concrete engineering problems. For example, Griffith's theory cannot predict the initiation of a crack; it can only predict the extension from an already existing defect. Furthermore, it can predict propagation, but gives no information on the direction and the possible bifurcation, branching, and coalescence. Although these issues are addressed within LEFM through ad hoc criteria, Griffith's criterion and LEFM remain cumbersome[61] where complex crack phenomena take place.

Francfort and Marigo[26] proposed to reformulate Griffith's criterion as a variational problem where we search for the displacement $\boldsymbol{u}$ and crack $\Gamma_c$ minimizing the body's total energy

$$\mathscr{E}(\boldsymbol{u}, \Gamma_c) = \int_{\Omega \backslash \Gamma_c} \Psi(\boldsymbol{u}) \mathrm{d}\Omega + \int_{\Gamma_c} G_c \mathrm{d}s - \int_{\Gamma_N} \boldsymbol{t} \cdot \boldsymbol{u} \mathrm{d}s, \tag{5}$$

where the second term is the crack energy. The associated variational problem is written as

$$\boldsymbol{u}, \Gamma_c = \operatorname{argmin}(\mathscr{E}(\boldsymbol{u}, \Gamma_c)) \text{ s.t. } \dot{\Gamma}_c \geq 0, \tag{6}$$

The $\dot{\Gamma}_c \geq 0$ constraint, with $\dot{\Gamma}_c = \frac{\mathrm{d}\Gamma_c}{\mathrm{d}t}$, is introduced to ensure non-receding cracks. With the fracture process formulated as Eq. 6, the initiation, propagation, and direction are all direct solutions to the minimization problem.

Although mathematically robust, the handling of unknown crack surfaces is difficult with common numerical methods like the finite element method. Therefore, ref. 27. proposed to approximate the crack using a continuous auxiliary field $d : \Omega \to [0,1]$, called the crack phase-field, through an elliptic regularization functional $\gamma(d)$. Here $d = 0$ describes an intact material whereas $d = 1$ describes a fully broken material point. Consequently, the surface integral over $\Gamma_c$ is replaced by the integral of $\gamma(d)$ over the body's volume. Using this approximation, and neglecting the external loads for the sake of brevity, the body's total energy reads

$$\mathscr{E}_d(\boldsymbol{u}, d) = \int_{\Omega} g(d) \Psi(\boldsymbol{u}) \mathrm{d}x + \int_{\Omega} G_c \gamma(d) \mathrm{d}x, \tag{7}$$

where $g(d)$ is the degradation function modulating the local stiffness. It is usually chosen so that $g(0) = 1$ and $g(1) = 0$, corresponding to the original stiffness of the intact material and zero stiffness of a fully broken material. Following the regularization, the variational problem becomes

$$\boldsymbol{u}, d = \arg\min(\mathscr{E}_d(\boldsymbol{u}, d)) \text{ s.t. } \dot{d} \geq 0. \tag{8}$$

Introducing pseudo time steps with $\Delta t^{n+1} = t^{n+1} - t^n$, the $\dot{d} \geq 0$ constraint can be rewritten as $d^{n+1} \geq d^n$. Consequently, the incremental form of Eq. 8 can be written as

$$\boldsymbol{u}^{n+1}, d^{n+1} = \arg\min(\mathscr{E}(\boldsymbol{u}^{n+1}, d^{n+1})) \text{ s.t. } d^{n+1} \geq d^n, \tag{9}$$

with

$$\mathscr{E}_d(\boldsymbol{u}^{n+1}, d^{n+1}) = \int_{\Omega} g(d^{n+1}) \Psi(\boldsymbol{u}^{n+1}) \mathrm{d}x + \int_{\Omega} G_c \gamma(d^{n+1}) \mathrm{d}x. \tag{10}$$

Through the crack phase-field, the continuity of the displacement field is preserved, and the minimization problem can easily be solved using a classical FEM. Since they rely only on potential energy minimization, phase-field models can predict crack initiation, propagation, and coalescence without any ad hoc criteria. Furthermore, and contrarily to well-known fracture theories like LEFM and Cohesive Zone Models (CZMs), they do not require augmented finite element or adaptive meshing techniques[62]. Additional information concerning the development of PFMs and their governing equations can be found in references[62–66]. A comparison with phase-field models used in interface problems can be found in refs. 67,68. Under the assumption of a perfectly brittle medium with isotropic elasticity, we adopt the simple and widely adopted AT1 model[69], where $g(d) = (1 - d)^2$ and $\gamma(d) = \frac{3}{8}(\frac{d}{l_c} + l_c|\nabla d|^2)$[70], where $l_c$ is a parameter controlling the width of the regularization. Furthermore, an elastic energy decomposition is needed to avoid crack propagation under compressive loads. Here, we use the spheric-deviatoric split, originally proposed by Amor et al.[71], where $\Psi = \Psi_+ + \Psi_-$ with

$$\Psi_+(\boldsymbol{\varepsilon}) = \frac{1}{2} K \langle \operatorname{tr}(\boldsymbol{\varepsilon}) \rangle_+^2 + \mu(\boldsymbol{\varepsilon_D} : \boldsymbol{\varepsilon_D}), \tag{11}$$

and

$$\Psi_-(\boldsymbol{\varepsilon}) = \frac{1}{2} K \langle \operatorname{tr}(\boldsymbol{\varepsilon}) \rangle_-^2. \tag{12}$$

$K = \lambda + \frac{2\mu}{3}$, where $\lambda$ and $\mu$ are the Lamé coefficients, and $\langle \cdot \rangle_\pm$ is the positive or negative ramp function. $\boldsymbol{\varepsilon_D}$ is the deviatoric part of the strain, with $\boldsymbol{\varepsilon_D} = \boldsymbol{\varepsilon} - \frac{1}{3} \operatorname{tr}(\boldsymbol{\varepsilon}) \boldsymbol{I}$ and $\boldsymbol{I}$ the second-order identity. Therefore, the fracture phase-field model reads as

$$\mathscr{E}_d(\boldsymbol{u}, d) = \int_{\Omega} \left( (1 - d)^2 \Psi_+(\boldsymbol{\varepsilon}) + \Psi_-(\boldsymbol{\varepsilon}) \right) \mathrm{d}x + \frac{3}{8} \int_{\Omega} G_c \left( \frac{d}{l_c} + l_c |\nabla d|^2 \right) \mathrm{d}x. \tag{13}$$

To study the fracture behavior of the patterned sheets, we suppose that the cells are homogeneous regions with an isotropic elastic and brittle fracture behavior, joined together by a thin second phase. To focus solely on the effect of the pattern type, the interfaces are also considered isotropic with a brittle behavior. Therefore, both the cells and the interfaces are modeled using Eq. 13, but they are given different material properties, similar to the method for heterogeneous material adopted by Hansen-Dörr et al. (2019)[72].

The PFM of Eq. 13 was implemented using the finite element method. Both the displacement field and the crack phase-field were discretized using bilinear quadrilateral elements. The minimization problem was solved using a modified Newton solver with an energy line search[28]. All geometries were meshed with a local refinement of $h = l_c/4$ in the crack region to avoid numerical overestimation of the toughness while maintaining a reasonable number of degrees of freedom.

Model Eq. 13 was applied to 2D compact-tension specimens with the same geometry as the acrylic specimens (Supplementary Fig. 1A, B). Brick-shaped and wavy cell patterns were implemented in the simulations to the compact-tension specimens and the absence of a pattern constituted the control. The cells were all given Young's modulus of $E^{cell} = 2000$ MPa, a Poisson's ratio of $\nu^{cell} = 0.35$, and a critical energy release rate of $G_c^{cell} = 0.3$ N/mm. The length scale was set

to $l_c^{cell} = 0.21$ mm to recover an ultimate strength of $\sigma_c^{cell} = 35$ MPa using the solution for a 1D bar under tension[69]. Since the objective here was not to predict or reproduce the behavior of a well-characterized material, but rather to study the effect of introducing interfaces of different shapes, the choice of these material parameters is rather arbitrary. Here, they were chosen to approximately reproduce the elastic modulus and peak load observed on the acrylic sheets fracture tests (Fig. 2E). For all numerical experiments, the interfaces were given a constant thickness of $T = 4l_c^{cell}$. Since $4l_c^{cell}$ is the width of the smeared crack with the *AT1* model[69], a propagating crack can be fully contained within the interface to recover an effective energy release rate $G_c^{eff} = G_c^{int}$.

In the first numerical experiment, we studied the effect on the material's effective toughness when all interface material parameters were assigned values equal to the cells, except for a toughness of $G_c^{int} = G_c^{cell}/2$. Figure 3A and Supplementary Fig. 2 present the force-displacement curves and the crack paths obtained for the control geometry and the three cell patterns. With the brick-shaped cells aligned longitudinally along the original crack path, the crack quickly finds an interface and follows it (Supplementary Fig. 2B). Consequently, the force reaction is smooth, but lower than the control geometry since the toughness is smaller. With the brick-shaped cells aligned transversely, the crack propagates horizontally, crossing interfaces with little deviation (Supplementary Fig. 2C). However, when an interface is parallel to the propagation direction and close to the crack path, bifurcations of the cracks are observed, similar to PMMA fracture tests (Fig. 2D). These bifurcations imply the need for a higher load to maintain crack propagation, which is observable as small peaks in the reaction curve (Fig. 3A). Because most of the propagation takes place within cells, the toughness of the transversely aligned brick cell pattern is similar to the control specimen. For puzzle-like cell patterns, the crack is deviated away from its expected straight-line propagation by the interfaces (Supplementary Fig. 2D). The deviation from the straight-line path creates the need for a higher load to drive crack propagation, which is reflected in the force-displacement curve (Fig. 3A).

In a second numerical experiment, we investigated the effect of the patterns when combined with both lower toughness and elastic modulus. Therefore, the interface material parameters were set equal to the cells, except for a toughness $G_c^{int} = G_c^{cell}/2$ and Young's modulus of $E^{int} = E^{cell}/2$. Figure 3 presents the obtained crack paths and force-displacement curves. Once again, the reaction force obtained with the longitudinally aligned brick-shaped cells is mostly smooth (Fig. 3B) since the crack remains within the interfaces (Fig. 3C). Consequently, with the toughness of the interfaces lower than that of the cells, the force-displacement curve obtained for this case is lower than the control specimen (Fig. 3B, F). With the brick-shaped cells aligned transversely (Fig. 3D), the reaction force oscillates in a decreasing sawtooth fashion (Fig. 3B). This is due to the difference in Young's modulus between the two phases, forcing the crack to re-initiate in the stiffer cells when arriving from the weak interfaces[30]. To re-initiate, the load must temporarily increase for the crack tip to accumulate the necessary strain energy, explaining the observable sawtooth behavior. The same phenomenon can be seen with the puzzle-like cell patterns obtained from *Arabidopsis* images. However, the crack remains longer within the interfaces since the latter are closer to the propagation direction (Fig. 3E). Consequently, fewer peaks are observed than with the transverse onion cells. Nevertheless, the inclusion of weak interfaces at angles other than parallel to the propagation direction augmented the effective toughness of the specimens with both the transverse onion cells and the wavy cells when compared to the control specimen.

The proposed PFM constitutes a simplification of the complex nature of the mechanical behavior of the *Arabidopsis* and onion cell interfaces. Nevertheless, varying interface toughness and stiffness in this PFM revealed that, for identical mechanical properties, the apparent toughness of the overall epidermal tissue is highly dependent on the cell organization/pattern. Furthermore, the numerical investigation showed that weak interfaces organized in a pattern deviating the crack away from its expected path can enhance the effective toughness of a material. The choice of an interface model and its associated parameters for epidermal tissue could be the subject of a complete investigation, as carried out in[72–75].

### Reporting summary

Further information on research design is available in the Nature Portfolio Reporting Summary linked to this article.

## Data availability

All data are available in the main text or the supplementary information. Source data are provided with this paper.

## Code availability

The phase-field fracture model codes are available at: https://github.com/ollamc/pfm-plantleafepidermis with https://doi.org/10.5281/zenodo.10056393.

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

## Acknowledgements

Sample preparation and image acquisition were performed at the McGill University Multi-Scale Imaging Facility, Sainte-Anne-de-Bellevue, Québec, Canada. We thank Philip M. Lintilhac (The University of Vermont) for lending us his tensile testing device. We also thank those colleagues who have provided feedback during the progress of this work. Specifically, we thank Dr. M. Shafayet Zamil for discussions and inspiration. Dr. Bara Altartouri provided helpful discussions during this study and kindly provided the Supplementary Fig. 3D micrograph. We would also like to extend our appreciation and apologies to researchers whose relevant work could not be cited due to space constraints. A.G. research team was supported by Natural Sciences and Engineering Research Council of Canada (NSERC) Discovery grant and Canada Research Chairs Program.

## Author contributions

A.J.B. conceptualized the study. A.J.B., O.L., F.P.G. and A.G. contributed to methodological design. A.J.B. designed and executed the fracture experiments, analyzed the data, and generated the graphs, O.L. conducted numerical simulations of the fracture and created the corresponding plots. A.J.B. and A.G. drafted the initial manuscript. A.J.B., O.L., F.P.G. and A.G. all participated in the writing and editing of the final manuscript. A.G. secured funding.

## Competing interests

The authors declare no competing interests.
