## [Peer Review File · Nature Communications]

Cell Geometry Regulates Tissue FractureReviewer #1 (Remarks to the Author):

Referee report on "Microscale geometrical features in the plant leaf epidermis confer enhanced resistance to mechanical failure" by Bidhendi et al.

The authors consider the influence of individual cells' geometry on the fracture of the epidermis of certain leaves. In particular, they compare the difference between tissues comprised of cells with strongly undulating boundaries with those comprised of cells with more straight boundaries. Striking differences are found in the fracture behavior in all three models: plant leaves, textured PMMA sheets, and numerical simulation.

The article is generally well written and I would consider the results quite interesting.

There is one thing that should be looked at more carefully. On p4 it is stated that the difference in Young's modulus requires a re-initiation of the crack in the stiffer tissue. There are, however, a number of other mechanisms that lead to a crack being arrested at material interfaces. See, e.g., [1]. Some investigation which mechanisms are relevant in plant leaf epidermal tissues would be very useful (using the in silico model to distinguish different effects). This would also strengthen the role of numerical simulation in this article, which now mostly just serves to replicate the experimental results.

Some minor issues:

- Fig. 2 has plots for 'Wavy-90' which is not explained - I assume it corresponds to different tear paths.
- Eqs (5,6) in the supplement are somewhat mixed between energy minimization and gradient flow. It might be better to use an incremental formulation here.
- For Arabidopsis no work of fracture is shown, is there a reason?
- I would think it is more 'geometry' than 'topology' (for the section title on p3).

[1] C-J. Hsueh et al., Stress fluctuation, crack renucleation and toughening in layered materials, JMPS (2018)

Reviewer #2 (Remarks to the Author):

Please see the attached word file.

Reviewer #2 Attachment on the following page

In this study, the authors perform fracture experiments to show the functional importance of the wavy pattern of the plant epidermis cell. They first produced a physical model to reproduce crack formation and undertook numerical simulation to reproduce the physical model. They also undertook tear testing. Finally, they listed special characteristics of crack propagation near the cell-cell boundary and observed the pattern in vivo. Although the author's intention is interesting, several points should be cleared, and biological significance should be reinforced.

Major issues

1. Is microfracture observed in physiological/pathological situations?

Under the current form, it is difficult to assess whether the mechanism is working in a natural situation. It is even unclear whether this kind of "crack" exist in a natural situation. The authors should show whether plant leaves in a natural situation. There are some cases where cell walls do not show winding structure, so the authors should assess whether the species with regular structure is vulnerable to such wound "in a natural situation".

2. Eq.11 in the supplement: "Phase-field"

2-1. It is interesting to see the phase field model, which is utilized to cell shape change or multicellular tissue patterning in mathematical biology is utilized in crack formation. The formalization is very different from phase-field we known (Allen-Cahn equation), so provide some explanation to compare the double well type phase field. It takes some time for me to understand the model with help of external source like

<https://pages.nist.gov/pfhub/docs/PhaseFieldVIII-Bourdin.pdf>

2-2. Supporting file p5, "well-known $A77$ model " – this model is not well known outside the author's community. Please include the reference.

2-3. The numerical simulation results in Fig.3: the results seem to be combination of line objects (attached figure) while actual simulation result should be continuous distribution.

It is more helpful to directly show d or \mathbf{u} .

3. Fig. 4. Was the two-leg trouser tear test done only in the onion epidermis? If so, describe the reason why the authors did not do this test on the wavy epidermis.

Minor issues:

1. Relationship with mechanistic models: there are several types of theoretical models for the mechanism of pattern formation. Some utilize buckling instability (Higaki, Takigawa-Imamura, et al., 2016) (Sapala et al., 2018) while others utilize localized cell wall production-degradation (Higaki, Kutsuna, et al., 2016). The authors should discuss the relationship between functional and mechanistic models.
2. Fig 4A. It is hard to understand the two-leg trouser tear test from the upper panel. I looked up other articles and understand the model, but I am not sure whether the test the authors have done is the same as the example below.

Download : Download full-size image

Fig. 2. The trousers test specimen.

10.1016/j.ijsolstr.2011.09.018

3. Fig. 5,6: I understand the characteristics the authors described in these figures are not considered in the previous numerical simulations. In this form, it is difficult to match which crack belongs to which example in Fig. 5A. I suggest the authors merge Figs. 5 and 6 and clarify which crack in Fig. 6 corresponds to Fig 5A,B, C.

References:

Higaki, T., Kutsuna, N., Akita, K., Takigawa-Imamura, H., Yoshimura, K., & Miura, T. (2016). A

Theoretical Model of Jigsaw-Puzzle Pattern Formation by Plant Leaf Epidermal Cells.

PLOS Computational Biology, 12(4), e1004833.

<https://doi.org/10.1371/journal.pcbi.1004833>

Higaki, T., Takigawa-Imamura, H., Akita, K., Kutsuna, N., Kobayashi, R., Hasezawa, S., & Miura,

T. (2016). Exogenous Cellulase Switches Cell Interdigitation to Cell Elongation in an RIC1-

dependent Manner in *Arabidopsis thaliana* Cotyledon Pavement Cells. *Plant and Cell*

Physiology, 58(1), pcw183. <https://doi.org/10.1093/pcp/pcw183>

Sapala, A., Runions, A., Routier-Kierzkowska, A. L., Gupta, M. Das, Hong, L., Hofhuis, H., Verger, S., Mosca, G., Li, C. B., Hay, A., Hamant, O., Roeder, A. H. K., Tsiantis, M., Prusinkiewicz, P., & Smith, R. S. (2018). Why plants make puzzle cells, and how their shape emerges. *ELife*, 7, 1–53. <https://doi.org/10.7554/eLife.32794>

Reviewer #3 (Remarks to the Author):

This manuscript tests the hypothesis that whether the wavy pattern of dicotyledon epidermal pavement cells confers resistance to tissue.

This hypothesis was posed previously and is probably the main hypothesis to explain WHY such a wavy pattern exists. Please, provide appropriate citations. However, this article goes beyond proposing a hypothesis; the authors test this hypothesis through numerical simulation and experimentation on 1) an artificial tissue layer formed of either wavy or hexagonal cells, and 3 types of biological tissue 2) Arabidopsis cotyledons and a tomato leaf presenting wavy cells on leaf epidermis and onion cells with hexagonal cells. The authors compare observations with modeling using a crack propagation theory. This is a very interesting and original work.

minor suggestions:

- 1) Lines 93-95 explain here PMMA and PFA frameworks. This is a non specialist journal. Readers need explanations accessible for non-specialists.
- 2) Define cohesive and adhesive failure (line 163). Non-specialist readers may have no idea what is it.
- 3) lines 134-135: define two-leg trousers and notched tear tests.
- 4) 187-188, maybe to this example add also trichrome maturation? This could be an interesting system to apply the crack propagation theory to explain cell separation.
- 5) Line 191-192: the emergence of a wavy pattern in the leaf epidermis is well studied. Finishing the article with this sentence gives the impression nobody ever studied it. Maybe cite different (divergent) views, beside those already provided.

The main weakness of this publication is a lack of transparency in the modeling part. Without access to code, it is not only impossible to recapitulate the results but also to fully judge the article.

Please provide the code. This publication is interesting and can be useful for the broad public as long as the code is published.

We thank the reviewers for their constructive comments that helped us improve the manuscript. Below you find point-by-point responses to all comments we received. The corresponding changes in the manuscript are highlighted in yellow.

Reviewer #1

The authors consider the influence of individual cells' geometry on the fracture of the epidermis of certain leaves. In particular, they compare the difference between tissues comprised of cells with strongly undulating boundaries with those comprised of cells with more straight boundaries. Striking differences are found in the fracture behavior in all three models: plant leaves, textured PMMA sheets, and numerical simulation.

The article is generally well written and I would consider the results quite interesting.

We appreciate these comments, thank you.

There is one thing that should be looked at more carefully. On p4 it is stated that the difference in Young's modulus requires a re-initiation of the crack in the stiffer tissue. There are, however, a number of other mechanisms that lead to a crack being arrested at material interfaces. See, e.g., C-J. Hsueh et al., Stress fluctuation, crack renucleation and toughening in layered materials, JMPS (2018).

Thank you for guiding us to other possible mechanisms. We have now included citation of this paper (pages 4/5).

Some investigation which mechanisms are relevant in plant leaf epidermal tissues would be very useful (using the in silico model to distinguish different effects). This would also strengthen the role of numerical simulation in this article, which now mostly just serves to replicate the experimental results.

Thank you for your thoughtful suggestion. Regarding the potential enhancement of our manuscript through further computational modeling, we concur that additional simulations could deepen the analysis and a parametric study can also shed light on behavior of plants with different leaf cell geometries. The suggested avenues represent entire projects in their own right. We appreciate the suggestion and intend to pursue this line of investigation in future studies.

Some minor issues:

- Fig. 2 has plots for 'Wavy-90' which is not explained - I assume it corresponds to different tear paths.

Thank you for pointing this out. Explanations have been added in the figure legend for clarity. The legend now reads: *“Wavy and Wavy-90 refer to a pavement cell pattern and its 90-degree rotation.”*

- Eqs (5,6) in the supplement are somewhat mixed between energy minimization and gradient flow. It might be better to use an incremental formulation here.

Thank you for the suggestion. We added details and an incremental formulation to clarify (please see Eq. 9 and 10 in the Supplementary material).

- For Arabidopsis no work of fracture is shown, is there a reason?

The absence of quantitative fracture data on Arabidopsis epidermis is related to the properties of this plant. The extreme delicacy of this leaf (compared to onion) made it impossible (in our hands) to isolate sufficiently large areas of intact epidermal tissue layers from underlying mesophyll layers. This limitation was among the reasons that prompted us to explore physical models made from plexiglass. We added this explanation on page 5.

- I would think it is more 'geometry' than 'topology' (for the section title on p3).

Agreed. The subheading was changed.

Reviewer #2:

In this study, the authors perform fracture experiments to show the functional importance of the wavy pattern of the plant epidermis cell. They first produced a physical model to reproduce crack formation and undertook numerical simulation to reproduce the physical model. They also undertook tear testing. Finally, they listed special characteristics of crack propagation near the cell-cell boundary and observed the pattern in vivo. Although the author's intention is interesting, several points should be cleared, and biological significance should be reinforced.

Major issues

1. Is microfracture observed in physiological/pathological situations?

Under the current form, it is difficult to assess whether the mechanism is working in a natural situation. It is even unclear whether this kind of “crack” exist in a natural situation. The authors should show whether plant leaves in a natural situation. There are some cases where cell walls

do not show winding structure, so the authors should assess whether the species with regular structure is vulnerable to such wound “in a natural situation”.

Thank you for pointing out that we could have been more elaborate in the ecological justification for the mechanism. We have now added references in the Introduction citing the different types of damage that that can be observed at the leaf surface due to abiotic and biotic causes (page 2). The passage reads '*This optimization of organ shape for photosynthesis comes at an ecological cost, however—high exposure to and susceptibility to mechanical damage by biotic and abiotic factors such as herbivory, pathogens, hail, sand storms and wind*^{9–12}. *The abrasive, piercing or slicing actions of these agents can initiate holes in the leaf surface that are prone to propagate as cracks.*'

2. Eq.11 in the supplement: “Phase-field”

2-1. It is interesting to see the phase field model, which is utilized to cell shape change or multicellular tissue patterning in mathematical biology is utilized in crack formation. The formalization is very different from phase-field we known (Allen-Cahn equation), so provide some explanation to compare the double well type phase field. It takes some time for me to understand the model with help of external source like <https://pages.nist.gov/pfhub/docs/PhaseFieldVIII-Bourdin.pdf>

Thank you for bringing this up. The phase-field method for fracture stems from the seminal work of Bourdin, Francfort, and Marigo (*Reference 9 in the supplementary material*) published in 2000, where the elliptical functional proposed by Ambrosio and Tortorelli (1992) was used to approximate the crack surface appearing in Griffith’s theory. Griffith’s theory, dating back to 1921, is the cornerstone of modern fracture mechanics. The phase-field method for fracture was subsequently applied to a wide range of problems. Numerous review papers, books, and chapters on the topic are available (*DOI: 10.1016/bs.aams.2019.08.001, 10.1007/s10659-007-9107-3, 10.1515/9783110497397, 10.1007/s00466-014-1109-y, 10.1016/j.engfracmech.2022.108234*).

Following your recommendation, we added references detailing the development of the phase-field fracture model (page 5 of the Supplemental material). Since the authors are familiar with the existence of the phase-field method to solve interface problems in other fields, such as multi-phase flow and solidification problems, but not experts, we also added references where a comparison with other formulation is offered.

To briefly answer your question regarding the double-well formulation, the latter is usually adopted since it allows a material point to reach two different local minima. For example, in solidification problems, a material point can reach the equilibrium states of solid or liquid (Gomez, Hector, and Kristoffer George van der Zee. "Computational phase-field modeling." 2017.). However, in phase-field fracture models, the degradation function $g(d)$ is used simply to modulate the stiffness of a material point, from intact ($d = 0$) to fully damaged ($d = 1$). Different shapes can be used for $g(d)$ depending on the material behavior, but the double-well formulation is rarely adopted since there is

no need for an equilibrium state somewhere between intact and damaged (DOI: 10.1016/bs.aams.2019.08.001).

2-2. Supporting file p5, “well-known AT1 model “ – this model is not well known outside the author’s community. Please include the reference.

Thank you for the suggestion. A reference for the AT1 model was added.

2-3. The numerical simulation results in Fig.3: the results seem to be combination of line objects (attached figure) while actual simulation result should be continuous distribution. It is more helpful to directly show d or \mathbf{u} .

Thank you for the comment. You are correct that the original results are indeed in form of continuous distributions. To highlight the crack path for a non-expert reader we have chosen to highlight the crack path as a line object. However, to address this comment, we have now added the original simulation results in the supplementary materials, Extended Data Fig. 4 and refer to it in the relevant figure legends.

3. Fig. 4. Was the two-leg trouser tear test done only in the onion epidermis? If so, describe the reason why the authors did not do this test on the wavy epidermis.

As mentioned in the answer to a similar question by the other reviewer: The reason lies in the technical difficulties inherent to sample preparation and experimentation on this tissue. Isolating sufficiently big and intact epidermal tissue layers from the delicate leaves of Arabidopsis and tomato was attempted but turned out to be impossible in our hands. This was among the reasons that prompted us to explore physical models made from plexiglass. We added this explanation on page 5.

Minor issues:

1. Relationship with mechanistic models: there are several types of theoretical models for the mechanism of pattern formation. Some utilize buckling instability (Higaki, Takigawa-Imamura, et al., 2016) (Sapala et al., 2018) while others utilize localized cell wall production-degradation (Higaki, Kutsuna, et al., 2016). The authors should discuss the relationship between functional and mechanistic models.

Thank you for drawing attention to the relation between our work and existing mechanistic models of pattern formation in plant cells. Our research is primarily concerned with the functional implications of cellular geometries and does not directly address the underlying mechanistic models such as buckling instability or localized cell wall production-degradation. Nonetheless, we recognize the significance of these mechanistic approaches in offering a comprehensive perspective on cell shape formation.

In response to your comment and to similar feedback from another reviewer, we have incorporated a passage at the outset of our conclusion (page 7) to elaborate on both mechanistic models that seeks to explain the emergence of wavy pavement cells. This added paragraph is intended to bridge the gap between functional and mechanistic models and provide a more complete view of the subject matter. The paragraph reads: *“Over recent decades, substantial research has focused on 'how' pavement cells develop their wavy geometries and 'why' these forms exist. Much progress has been made regarding the 'how'—the developmental and mechanical mechanisms behind these shapes—through the investigation of biomechanical processes and the analyses of molecular players^{2,3,6,36-41}. The 'why'—the physiological or evolutionary benefits—on the other hand, remains poorly understood. Existing hypotheses suggest these geometries could either increase tissue elasticity or minimize stress within larger epidermal cells, but these notions have remained speculative.”*

2. Fig 4A. It is hard to understand the two-leg trouser tear test from the upper panel. I looked up other articles and understand the model, but I am not sure whether the test the authors have done is the same as the example below.

We have now provided a more detailed explanation of the two types of tests (page 5) and believe that they make the drawing easier to understand.

3. Fig. 5,6: I understand the characteristics the authors described in these figures are not considered in the previous numerical simulations. In this form, it is difficult to match which crack belongs to which example in Fig. 5A. I suggest the authors merge Figs. 5 and 6 and clarify which crack in Fig. 6 corresponds to Fig 5A,B, C.

The size of the two figures would have made it difficult to merge them without making the individual micrographs very small. Instead, we have now amended Figure legend 6 to specify which kind of crack propagation (as shown in Figure 5) is visible on these examples.

The following passage was added: *'The two-leg trousers tear test is a mechanical test setup where the sample is shaped like a pair of trousers with a pre-cut to initiate a tear; to measure tear resistance, the 'legs' are pulled apart in the same direction as the leg motions of a walking person. The notched tear test involves a pre-cut or notch in the sample, subjected to tensile force to assess resistance to tear propagation. Both tests are used to analyze fracture mechanics and tear resistance.'*

Reviewer #3:

This manuscript tests the hypothesis that whether the wavy pattern of dicotyledon epidermal pavement cells confers resistance to tissue.

This hypothesis was posed previously and is probably the main hypothesis to explain WHY such a wavy pattern exists. Please, provide appropriate citations.

We don't seem to have come across the publication the reviewer alludes to. Very well-cited articles propose roles for the undulating cell circumference in conferring elasticity to the epidermal tissue under tensile stress (Sotiriou et al. 2018) or in minimizing surface stress at increasing cell volumes (Sapala et al. 2018). We cite these in the Introduction. A renewed literature search did not yield any published hypothesis for the **prevention of crack-propagation as evolutionary advantage for leaf epidermis waviness**. We would much appreciate if we were guided to the actual article that the reviewer alludes to.

However, this article goes beyond proposing a hypothesis; the authors test this hypothesis through numerical simulation and experimentation on 1) an artificial tissue layer formed of either wavy or hexagonal cells, and 3 types of biological tissue 2) Arabidopsis cotyledons and a tomato leaf presenting wavy cells on leaf epidermis and onion cells with hexagonal cells. The authors compare observations with modeling using a crack propagation theory. This is a very interesting and original work.

Thank you, we appreciate the comment.

minor suggestions:

1) Lines 93-95 explain here PMMA and PFA frameworks. This is a non specialist journal. Readers need explanations accessible for non-specialists.

Agreed, we added explanations for these. For PMMA at its first occurrence (section Geometry influences fracture propagation). The sentence now reads as follows:
".....the cast polymethylmethacrylate (PMMA), commonly known as acrylic or Plexiglass, sheets with the shapes of epidermal..."

We believe the reviewer refers to PFM and not PFA as the latter does not appear in the text. The added explanation now reads as follows:

"PFM is a computational method used for simulating the formation and evolution of complex fractures in a material."

2) Define cohesive and adhesive failure (line 163). Non-specialist readers may have no idea what is it.

Thank you for pointing out the need for clarification on these technical terms. To assist non-specialist readers, we have included concise definitions for both "cohesive failure" and "adhesive failure". The amended sentence now reads:

"Whether the actual splitting at the middle lamella represented an adhesive failure (rupture at the interface between the bonding material and substrate) or a cohesive failure (rupture of the bonding material itself) (Fig. 5B) was impossible to determine."

3) lines 134-135: define two-leg trousers and notched tear tests.

Thank you for your suggestion to clarify the terminology. We have now included brief definitions for both "two-leg trousers tear test" and "notched tear test" in the text as highlighted in "Crack behavior in the plant epidermis validates deflection and meandering" section (page 5).

4) 187-188, maybe to this example add also trichome maturation? This could be an interesting system to apply the crack propagation theory to explain cell separation.

This suggestion is much appreciated. Depending on the geometry of the cell structure at the trichome base, as well as the potentially changing cell wall mechanics and adhesion properties at these locations, the fracture path would certainly be influenced and investigating this is a much appreciated suggestion. This avenue is beyond the scope of the present study, however.

5) Line 191-192: the emergence of a wavy pattern in the leaf epidermis is well studied. Finishing the article with this sentence gives the impression nobody ever studied it. Maybe cite different (divergent) views, beside those already provided.

Thank you for your comment. In response, we have added citations in the opening passage of the conclusion section (page 5), to more comprehensively refer to papers involving both 'how' and 'why' these cellular shapes emerge. Furthermore, divergent viewpoints on the 'why' have been included in the introduction to provide a balanced perspective.

The main weakness of this publication is a lack of transparency in the modeling part. Without access to code, it is not only impossible to recapitulate the results but also to fully judge the article.

Please provide the code. This publication is interesting and can be useful for the broad public as long as the code is published.

Agreed. We have now uploaded the code on GitHub: <https://github.com/ollamc/pfm-plantleafepidermis>

This is now explicitly stated in the Data and materials availability section.

Reviewer #2 (Remarks to the Author):

I think the authors have corrected the manuscript appropriately.

Reviewer #3 (Remarks to the Author):

The authors have addressed my concerns. I have no further questions.